# Developing and validating a frailty score based on patient-reported outcome 3 months after stroke: A Riksstroke-based study

**Joakim Wallmark**📵*, **Marie Wiberg**📵, **Marie Eriksson**📵

Department of Statistics, Umeå School of Business, Economics and Statistics, Umeå University, Umeå, Sweden

* joakim.wallmark@umu.se

## Abstract

**Background:** Frailty is common after stroke and linked to poor outcomes, but many measures are clinician-rated, time-consuming, and not suited to patient-reported data. To address these issues, we developed and validated a frailty score from the Swedish Stroke Register (Riksstroke) three-month follow-up questionnaire.

**Methods:** We analyzed responses from 19,470 stroke survivors to nine patient-reported items covering function, mood, fatigue, pain and general health, in the 2021–2022 Riksstroke questionnaire. Dimensionality was assessed with Mokken Scale Analysis and exploratory factor analysis. Item response theory (IRT) was used for score computation. Competing graded response IRT models (unidimensional, correlated-factor, bifactor) were compared, and measurement fairness was examined using differential item functioning (DIF) across age, sex, and education. Prognostic validity was tested with Kaplan–Meier curves and Cox regression for all-cause mortality.

**Results:** From the Mokken Scale Analysis, all items met scalability criteria. Factor analysis suggested two correlated interpretable facets (Physical Functioning; Well-being/Mental Health). A bifactor IRT model provided the best fit to the data, comprising a general frailty dimension while addressing the strong correlation between the facets. DIF was minimal for sex and education, with modest age-related effects. Higher frailty scores were associated with increased mortality in adjusted Cox models and Kaplan–Meier curves. Tools for computing frailty scores are available at https://github.com/joakimwallmark/frailty-irt-scores.

**Conclusions:** A robust, fair, and prognostically meaningful frailty score can be derived from patient-reported items in Riksstroke. More broadly, the study demonstrates how routinely collected patient-reported outcome measures can be leveraged to build scalable frailty scores, offering efficient cost-effective tools for monitoring outcome and guiding quality improvement in stroke care.

**Data availability statement:** The data analyzed in the study consist of third-party data from Riksstroke and Statistics Sweden and their use has been approved by the Swedish Ethical Review Authority (document number 2023-07750-01). Because of the sensitive nature of the data and according to Swedish legislation (https://etikprovningsmyndigheten.se/for-forskare/vad-sager-lagen/) data cannot be made available for use beyond what has been approved by the ethical review authority. Therefore, the data cannot be made publicly available. Data may be made available from Riksstroke (contact via riksstroke@regionvasterbotten.se) or Statistics Sweden (contact via mikrodata@scb.se) upon reasonable request by researchers who meet the criteria for access to confidential data according to Swedish laws and regulations.

**Funding:** The study was funded by the Swedish Research Council, https://www.vr.se/english.html, (grant number 2024-02846 to ME, and grant number 2022-02046 to MW). The funders had no role in study design, data collection and analysis, decision to publish, or preparation of the manuscript.

**Competing interests:** No authors have competing interests.

## Introduction

Stroke is a leading cause of long-term disability and premature mortality worldwide [1]. In Sweden, the national stroke register (Riksstroke) issues a follow-up questionnaire three months after stroke to monitor patient recovery and to inform quality improvement at the local, regional, and national level [2]. A central challenge is how to summarize this patient-reported information into a single, interpretable measure that reflects overall health and can be used for risk stratification.

Frailty provides one such framework. Frailty is commonly defined as a clinical syndrome of reduced physiological reserve and resilience, in which seemingly minor stressors can cause disproportionate deterioration in health [3,4]. Conceptually, it reflects the erosion of the body's in-built reserves across multiple organ systems, making individuals more vulnerable to illness, complications, and mortality. In stroke, frailty is highly prevalent: systematic reviews estimate that about one quarter of patients presenting with acute stroke are already frail, and nearly half are frail or prefrail when broader definitions are applied [3,5,6]. Frailty is clinically important because it predicts worse recovery, reduced effectiveness of acute treatments, longer hospital stays, institutionalization, and higher mortality [5].

Frailty is not the same as chronological age or disability. A patient can be frail without being severely disabled, and vice versa [5]. Importantly, frailty measures have sometimes outperformed disability scales in predicting long-term outcomes [4]. This has led to repeated calls for standardized, explicit assessment of frailty in stroke research and clinical care, rather than assuming that age or disability adequately capture it [4,6].

Several approaches to measuring frailty are used in clinical and research settings. *Phenotype-based tools* such as the Fried criteria [7] or FRAIL scale [8] define frailty by meeting a set of physical criteria (e.g., weakness, slowness, exhaustion, low activity, weight loss). *Cumulative-deficit indices*, such as the Frailty Index (FI) [9], the electronic Frailty Index (eFI) [10], and the Hospital Frailty Risk Score (HFRS) [11], quantify frailty as the proportion of accumulated health deficits. In clinical practice, judgment-based tools such as the *Clinical Frailty Scale (CFS)* [12,13] are also widely used, providing a 9-point clinician-rated summary of overall fitness or frailty based on comorbidity, function, and cognition. Although these measures have strong prognostic validity for outcomes such as mortality and institutionalization, they are clinician-assessed, time-consuming, and not tailored to large-scale patient-reported follow-up data [14–16].

The Riksstroke follow-up collects 20 patient-reported items [17,18]. The English translation of the questionnaire is currently available at https://www.riksstroke.org/wp-content/uploads/2021/03/3-manaders-uppfoljning-2021_-engelska-1.pdf. In this study, we consider nine of these items that cover function, mood, fatigue, pain, and general health—domains that are central to frailty. Methodologically, frailty can be viewed as a *latent construct* underlying these observed indicators. Psychometric models, such as Item Response Theory (IRT) [19], provide a principled way to evaluate dimensionality, estimate fair and comparable scores across subgroups, and

validate them against external outcomes. Previous work on patient-reported outcome measures (PROMs) in stroke has applied IRT for scale development (but not specifically for frailty) [20] and shown the prognostic value of PROMs for future health events [21].

In contrast to clinician-rated frailty tools (e.g., HFRS, eFI), this study develops and validates an IRT-derived frailty score directly from patient-reported outcome items in Riksstroke. We (1) assess dimensionality using Mokken Scale Analysis and exploratory factor analysis, (2) compare unidimensional, correlated-factor, and bifactor graded IRT models, (3) examine and test measurement fairness across age, sex, and education using differential item functioning (DIF) analyses, and (4) evaluate external validity by linking the score to all-cause mortality.

## Material and methods

### Riksstroke

This study is based on data from Riksstroke, the national Swedish stroke register, which includes a structured follow–up questionnaire administered three months after stroke. The questionnaire includes patient–reported items that cover domains such as activities of daily living, mobility, mood, fatigue, pain, and general health. For the present analyses, we focused on a subset of nine items (English versions presented in Table 1) that map most directly onto established frailty domains. Items regarding transportation and cognitive symptoms (memory/concentration) were excluded a priori. This was done to minimize environmental confounding (transportation) and to focus the latent trait on physical resilience and general well-being, adhering closer to the physical frailty phenotype rather than a cumulative deficit index.

This study was approved by the Swedish Ethical Review Authority (reference number 2023-07750-01). Data were pseudonymized before it was submitted for the study. It was accessed for research December 5, 2024. Patients are informed about registration in the quality registry Riksstroke that the registry aims to support high and consistent quality of care for stroke patients throughout Sweden, and that data may be used for research purposes. In accordance with the Personal Data Act (Swedish law No. SFS 1998:204), no informed consent is needed to collect data from medical charts or inpatient records for quality registries. However, patients are informed of their rights to deny participation (opt-out consent).

The source population comprised all patients with ischemic or hemorrhagic stroke (International Classification of Diseases 10th Revision, ICD-10 codes I61, I63, and I64) during 2021–2022 who responded to the three–month follow–up questionnaire. We included all respondents with available answers to the nine frailty–related items. Questionnaires completed without the patient's involvement (i.e., filled in solely by a relative or healthcare professional) were excluded, as patient input is essential for measurement invariance analysis [22]. To preserve the ordinal structure of responses, "don't know" responses were coded as missing. As a pre–specified data–quality criterion, records with more than three missing responses across the nine items were excluded. The resulting analytic sample, along with descriptive characteristics, is reported in the Results section.

### Statistical methods

Item responses were treated as ordered as displayed in Table 1; we denote the response to item $j$ as $Y_j \in \{1, \dots, m_j\}$, and code "don't know" as missing.

**Dimensionality assessment.** To create a frailty score, the items must first be shown to measure a single, coherent underlying construct. We therefore assessed dimensionality with two complementary methods: Mokken Scale Analysis (MSA) [23,24] and an Exploratory Factor Analysis (EFA).

MSA was employed to assess the unidimensionality and hierarchical ordering of the items. MSA evaluates if items form a scale where individuals endorsing a more "difficult" or "extreme" item are also likely to endorse all "easier" or less "extreme" items. This involved an examination of Guttman errors to identify response patterns inconsistent with a Guttman-style ordering. Scalability was assessed using Loevinger's H coefficients, including item-pair ($H_{ij}$), individual item

**Table 1. The nine selected questionnaire items.** A short name (in bold) has been assigned to each item. Response options have been rearranged in the order from most to least favorable outcome

| No. | Question Text | Response Options |
|---|---|---|
| 1 | **Return to life/activities:** Have you been able to return to the life and activities you had before the stroke? | 1: Yes completely.<br>2: Yes but not quite as before.<br>3: No.<br>4: Don't know. |
| 2 | **Mobility:** How is your mobility now? | 1: I am able to move around both indoors and outdoors without the help of another person.<br>2: I am able to move around indoors but need help to move around outdoors.<br>3: I need help to move around both indoors and outdoors. |
| 3 | **Toilet help:** Do you need help when using the toilet? | 1: I can use the toilet without any help.<br>2: I need help when using the toilet. |
| 4 | **Dressing help:** Do you need help when dressing and undressing yourself? | 1: I can dress and undress myself without any help.<br>2: I need help to dress and undress myself. |
| 5 | **Dependent on support:** Are you currently dependent on support or help? | 1: No.<br>2: Yes partly.<br>3: Yes completely.<br>4: Don't know. |
| 6 | **Down/depressed/anxious:** Since your stroke, do you feel more downhearted/depressed or anxious than before? | 1: No.<br>2: Yes.<br>3: Don't know. |
| 7 | **General health status:** How would you rate your overall health? | 1: Very good.<br>2: Fairly good.<br>3: Fairly poor.<br>4: Very poor.<br>5: Don't know. |
| 8 | **Increased fatigue:** Since your stroke, do you feel more tired than before and does this affect your ability to carry out everyday activities? | 1: No.<br>2: Yes.<br>3: Don't know. |
| 9 | **New pain:** Since your stroke, are you experiencing any new types of pain? | 1: No.<br>2: Yes.<br>3: Don't know. |

($H_i$), and overall scale homogeneity ($H_S$) coefficients. Generally, scales are considered weak if $0.3 \leq H < 0.4$, moderate if $0.4 \leq H < 0.5$, and strong if $H \geq 0.5$, with individual items ideally having $H_i \geq 0.3$ [24]. Since MSA served as a descriptive check of scalability/monotonicity rather than as a basis for IRT model specification, it was conducted on the full dataset to maximize precision of Loevinger's H estimates.

EFA was performed to further identify the underlying latent constructs, and to empirically establish item–factor mappings for IRT model specification. Given the ordinal nature of the item responses, a polychoric correlation matrix was used. To reduce bias from discovering the latent structure and later constructing IRT models using the same data, a split-sample workflow was employed. The EFA was conducted using a randomly selected 20% subset, leaving the remaining 80% as a confirmation subset for IRT model fitting. The suitability of the EFA subset for factor analysis was assessed using the Kaiser-Meyer-Olkin (KMO) measure [25]. The number of factors to retain was guided by multiple criteria including Kaiser's criterion (eigenvalues > 1), scree plot examination [26], and parallel analysis [27]. EFA was performed using Minimum Residual (minres) estimation with an oblique (oblimin) rotation to allow factors to correlate.

**Item response modeling and score scale construction.** Once the dimensional structure was understood from MSA/EFA, we modeled item responses with graded response models (GRM) [28]. Guided by the EFA, we conducted confirmatory model selection on the remaining 80% confirmation subset, comparing: (i) a unidimensional GRM, (ii) a correlated-factor confirmatory GRM with two correlated factors ('Physical Functioning' and 'Well-being and Mental

Health'), and (iii) a bifactor GRM with one general frailty factor $\theta$ (common to all items) and two orthogonal specific factors ($\theta_{s1}, \theta_{s2}$) corresponding to Physical Functioning and Well-being/Mental Health.

Model fit for all confirmatory models was assessed using the Root Mean Square Error of Approximation (RMSEA), the Standardized Root Mean Square Residual (SRMSR), the Comparative Fit Index (CFI), and the Tucker-Lewis Index (TLI) [29,30]. After exploring different IRT structures, the selected models were refitted using the full dataset to obtain factor scores for DIF and survival analyses. All IRT models were fitted using the `mirt` package [31].

For transparency and later parameter interpretation, the functional form of the bifactor model is given below. Let $\theta$ denote the *general frailty factor* (loaded by all items), and $\theta_{s1}$ and $\theta_{s2}$ represent orthogonal specific factors for *Physical Functioning* and *Well-being/Mental Health*, respectively. For item $j$, the graded response model specifies the cumulative item score probabilities

$$\Pr(Y_j \geq k \mid \theta, \theta_{s1}, \theta_{s2}) = \left(1 + \exp(a_{jg}\,\theta + a_{js1}\,\theta_{s1} + a_{js2}\,\theta_{s2} + d_{jk})\right)^{-1}, \tag{1}$$

where $a_{jg}, a_{js1}, a_{js2}$ are item slopes (discriminations) on the general/specific factors, and $d_{jk}$ are ordered intercept ($d_{j1} > \cdots > d_{j,m_j-1}$). Item score probabilities follow from adjacent differences

$$\Pr(Y_j = k \mid \theta, \theta_{s1}, \theta_{s2}) = \Pr(Y_j \geq k \mid \theta, \theta_{s1}, \theta_{s2}) - \Pr(Y_j \geq k+1 \mid \theta, \theta_{s1}, \theta_{s2}),$$

with the convention $\Pr(Y_j \geq 1 \mid \theta, \theta_{s1}, \theta_{s2}) \equiv 1$. All three factors were assumed orthogonal. Slopes on non-applicable specific factors were fixed to 0.

**Interpreting slopes and intercepts.** In Eq (1), the vector of slopes for item $j$ is $\mathbf{a}_j = (a_{jg}, a_{js1}, a_{js2})$. Larger slopes indicate greater discrimination (steeper response curves). The ordered intercepts $d_{jk}$ determine where the cumulative curves switch along the latent dimensions; for a given score boundary $k$, the location along the item's most sensitive direction is $b_{jk}^* = -d_{jk}/\|\mathbf{a}_j\|$, where $\|\mathbf{a}_j\| = \sqrt{a_{jg}^2 + a_{js1}^2 + a_{js2}^2}$. Intuitively, $b_{jk}^*$ is the point on the latent continuum (projected onto $\mathbf{a}_j$) where the probability of responding with a score $\geq k$ is 0.5. Throughout, we report factor scores on the general frailty factor, scaled to mean 0 and variance 1 (higher = worse).

**Score fairness.** To ensure the frailty score is fair and comparable across different groups, we investigated whether items functioned differently for various subgroups of the population. A DIF analysis was conducted for three demographic variables: sex (male vs. female), education level (Primary, Secondary, and University), and age. For the latter, participants were dichotomized at the sample median of 76 years, such that individuals aged 76 years or below comprised the younger group and those older than 76 years comprised the older group. To identify DIF items and anchor items (items assumed free of DIF, required for model identification), we used the drop-sequential anchor-purification procedure based on sample-size–adjusted BIC [32] implemented in the `mirt` package [31]. To gauge the practical magnitude of DIF, we reported the Expected-Score Standardized Difference (ESSD) [33], a Cohen's *d*–style index. Following conventional thresholds [34], we interpreted |ESSD| $\approx$ 0.20, 0.50, and 0.80 as small, medium, and large DIF, respectively. All DIF analyses used models refit on the full dataset to maximize precision for group comparisons.

**External validity.** To evaluate the prognostic validity of the frailty scores, we examined their association with all-cause mortality. Individuals were tracked up to 1,100 days after completing the questionnaire. Latent trait estimates of frailty ($\hat{\theta}$) were obtained using expected a posteriori estimation [35] from the final IRT models refit on the full dataset. Score estimates were computed using both the bifactor model (general dimension $\theta$ estimates) and the unidimensional model.

For descriptive analysis, $\hat{\theta}$ was classified into quartiles (Q1–Q4) and Kaplan–Meier survival curves were estimated to visualize differences in survival between frailty levels. To formally quantify associations, we fitted Cox proportional hazards regression models with time-to-death as the outcome. We adjusted for age, sex, and education to reduce potential

confounding by demographic characteristics. In addition, we specified interaction terms between $\hat{\theta}$ and each of the covariates to test for potential effect modification, i.e. whether the strength of the association between frailty and mortality differs across age, sex, or educational groups. Continuous variables (age and $\hat{\theta}$) were mean-centered prior to modeling to facilitate interpretation of main effect regression coefficients.

## Results

A total of 22,389 three–month questionnaires were available from patients with ischemic or hemorrhagic stroke in 2021–2022. We excluded 2,531 questionnaires that were completed without the patient present (11.3%). After coding "don't know" as missing, a further 388 records (1.7%) were excluded due to having more than three missing responses across the nine items. The final analytic sample comprised 19,470 patients (87.0% of the starting cohort). Demographic characteristics of the analytic sample are presented in Table 2, and item–level response distributions are shown in Fig 1.

### Dimensionality

MSA conducted on the full dataset provided strong evidence that the nine items form a unidimensional and robust scale, with an overall Loevinger's $H_S$ of 0.614. All items and item pairs met scalability criteria based on the conventional lower bound threshold of 0.3. This suggests that the items can be conceptualized along a single latent continuum, where higher scores indicate a greater level of the underlying trait (worse recovery and overall well-being). Refer to Table 3 for results using different thresholds. 5.5% of the respondents had notable Guttman errors, suggesting that the hierarchical ordering of items holds for most individuals.

EFA was conducted on 20% of the data set to investigate the underlying factor structure of the nine questionnaire items. The KMO measure of sampling adequacy indicated that the data were suitable for factor analysis (KMO item-level scores ranged between 0.79 and 0.95). Based on multiple criteria, including the examination of scree plots, Kaiser's criterion (eigenvalues > 1), and parallel analysis, a correlated-factor solution was deemed most appropriate. This correlated-factor solution, derived using minimum residual estimation and an oblimin rotation, accounted for 76% of the total variance. Overall model fit indices were: RMSR = 0.013, TLI = 0.968, and RMSEA = 0.093 (90% CI [0.087, 0.099]).

The first factor explained 43% of the variance, and the second factor explained 33%. Factor loadings, communalities ($h^2$), uniquenesses ($u^2$), and item complexity (com) are presented in Table 4. Item communalities were generally high (ranging from 0.407 to 0.988). The mean item complexity of 1.13 indicated a relatively simple structure with items tending to load primarily on a single factor. The two factors exhibited a moderate positive correlation ($r = 0.57$). This suggests that there are two related but distinguishable content areas or facets within the items.

**Table 2. Summary statistics across demographics.**

| Characteristic | Hemorrhage N = 1,777 | Ischemic N = 17,567 | Unspecified N = 126 | Overall N = 19,470 |
|---|---|---|---|---|
| **Age** | | | | |
| Mean (SD) | 71 (13) | 74 (11) | 78 (10) | 74 (12) |
| Median [Q1, Q3] | 74 [63, 80] | 76 [68, 82] | 80 [74, 84] | 76 [68, 82] |
| **Sex** | | | | |
| Male | 1,028 (58%) | 9,799 (56%) | 62 (49%) | 10,889 (56%) |
| Female | 749 (42%) | 7,768 (44%) | 64 (51%) | 8,581 (44%) |
| **Education** | | | | |
| Missing | 20 (1.1%) | 180 (1.0%) | 0 (0%) | 200 (1.0%) |
| Primary | 469 (26%) | 5,101 (29%) | 49 (39%) | 5,619 (29%) |
| Secondary | 804 (45%) | 7,651 (44%) | 55 (44%) | 8,510 (44%) |
| University | 484 (27%) | 4,635 (26%) | 22 (17%) | 5,141 (26%) |

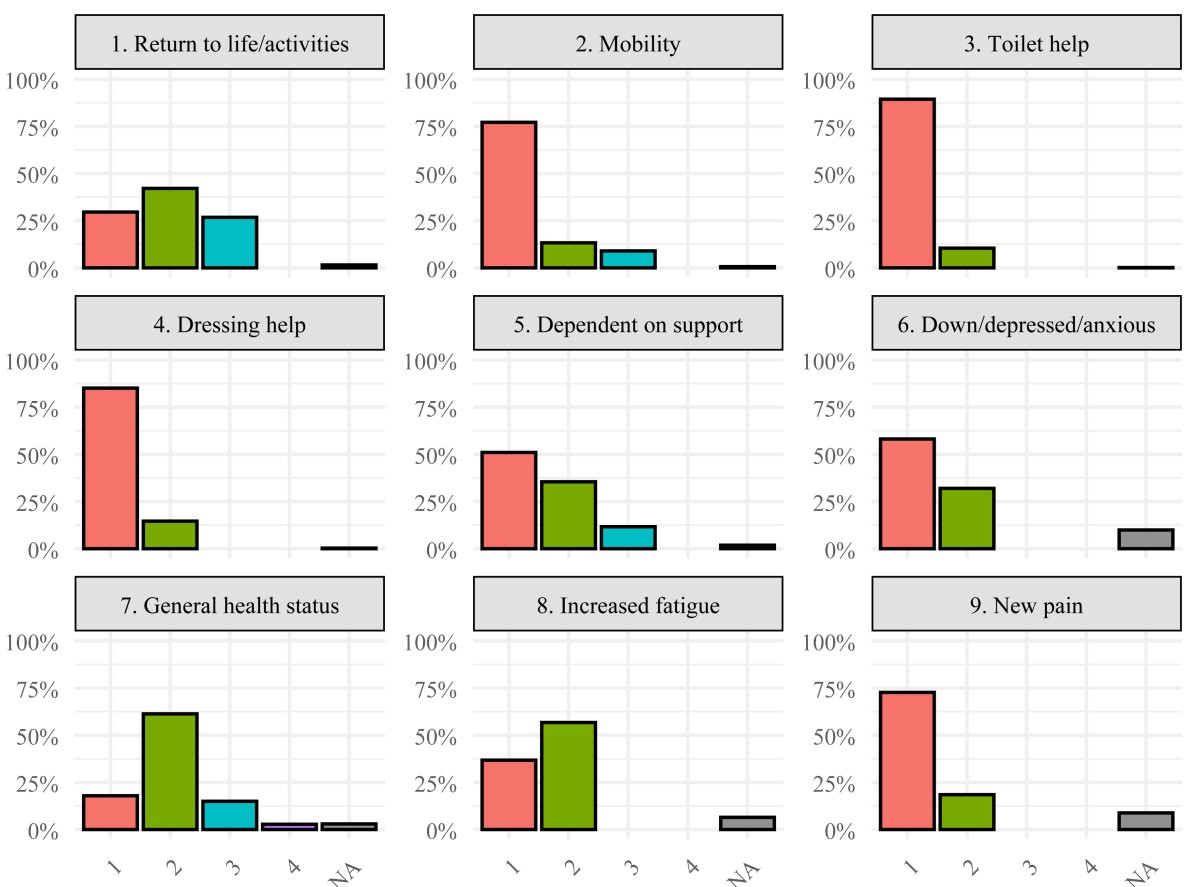

**Fig 1. Item response percentages.** NA indicates missing responses. Refer to Table 1 for item response descriptions of the digits for each item.

**Table 3**. **Loevinger's H coefficients and automated item selection procedure results.** H coefficient standard errors are shown in parentheses. Scalable items are denoted by 1 at a given threshold, while 0 indicates the item was unscalable.

| Item | $H_i$(SE) | Threshold | | | | | | | | |
|---|---|---|---|---|---|---|---|---|---|---|
| | | 0.20 | 0.25 | 0.30 | 0.35 | 0.40 | 0.45 | 0.50 | 0.55 | 0.60 |
| 1 | 0.672 (0.005) | 1 | 1 | 1 | 1 | 1 | 1 | 1 | 1 | 1 |
| 2 | 0.637 (0.005) | 1 | 1 | 1 | 1 | 1 | 1 | 1 | 1 | 1 |
| 3 | 0.688 (0.007) | 1 | 1 | 1 | 1 | 1 | 1 | 1 | 1 | 1 |
| 4 | 0.652 (0.006) | 1 | 1 | 1 | 1 | 1 | 1 | 1 | 1 | 1 |
| 5 | 0.629 (0.006) | 1 | 1 | 1 | 1 | 1 | 1 | 1 | 1 | 1 |
| 6 | 0.507 (0.008) | 1 | 1 | 1 | 1 | 1 | 1 | 1 | 0 | 0 |
| 7 | 0.641 (0.005) | 1 | 1 | 1 | 1 | 1 | 1 | 1 | 1 | 1 |
| 8 | 0.668 (0.007) | 1 | 1 | 1 | 1 | 1 | 1 | 1 | 1 | 1 |
| 9 | 0.401 (0.008) | 1 | 1 | 1 | 1 | 1 | 0 | 0 | 0 | 0 |

Factor 1 was primarily defined by items concerning mobility and activities of daily living (Item 2: Mobility; Item 3: Toilet help; Item 4: Dressing help; Item 5: Dependent on support) and was labeled *Physical Functioning*. Factor 2 was characterized by items related to mood, fatigue, pain, general health perception, and return to previous activities (Item 1: Return to life/activities; Item 6: Down/depressed/anxious; Item 7: General health status; Item 8: Increased fatigue; Item 9: New pain) and was labeled *Well-being and Mental Health*. Loadings larger than 0.2 are shown in Table 4.

**Table 4. Factor loadings (F1 and F2), Communalities ($h^2$), Uniquenesses ($u^2$), and Complexity (com) from EFA with Oblimin Rotation.** Loadings < 0.20 are omitted for clarity.

| Item | F1 | F2 | $h^2$ | $u^2$ | com |
|---|---|---|---|---|---|
| 1: Return to life/activities | 0.391 | 0.572 | 0.756 | 0.244 | 1.81 |
| 2: Mobility | 0.927 | | 0.898 | 0.102 | 1.00 |
| 3: Toilet help | 1.032 | | 0.988 | 0.012 | 1.01 |
| 4: Dressing help | 0.956 | | 0.907 | 0.093 | 1.00 |
| 5: Dependent on support | 0.789 | | 0.759 | 0.241 | 1.06 |
| 6: Down/depressed/anxious | | 0.812 | 0.611 | 0.389 | 1.01 |
| 7: General health status | 0.245 | 0.661 | 0.677 | 0.323 | 1.27 |
| 8: Increased fatigue | | 0.954 | 0.837 | 0.163 | 1.01 |
| 9: New pain | | 0.627 | 0.407 | 0.593 | 1.00 |

*Note:* Item descriptions are abbreviated.

### Item response modeling and score scale construction

Based on the results of the MSA and EFA analysis, the fit of several IRT models was compared, and the results are presented in Table 5. The models were fit on the 80% confirmation subset as previously explained.

The initial unidimensional IRT model, which posited that all items measure a single underlying construct, demonstrated a poor fit to the data. The RMSEA was 0.117 (90% CI [0.115, 0.120]), and the SRMSR was 0.094, both exceeding commonly accepted thresholds for good fit [36]. The CFI (0.936) and TLI (0.914) were also poor. These results strongly suggested that a multidimensional structure would better represent the data.

A confirmatory correlated-factor GRM was constructed based on the Physical Functioning and Well-being/Mental Health structure from the EFA. This model showed acceptable fit, with RMSEA = 0.059 (90% CI [0.056, 0.061]), SRMSR = 0.054, CFI = 0.984, and TLI = 0.978. The two latent factors were highly correlated ($r = 0.78$), suggesting a strong overarching general dimension.

Finally, a bifactor GRM was estimated on the 80% confirmation subset, including one general factor (all items) plus the two specific factors identified above. All factors were assumed uncorrelated. This model provided the best overall fit, with RMSEA = 0.026 (90% CI [0.023, 0.030]), SRMSR = 0.033, CFI = 0.998, and TLI = 0.996 (Table 5). The bifactor structure effectively addressed the high correlation between the CFA factors by modeling their shared variance through a dominant general dimension.

**Item-level interpretation.** Linking back to Eq (1), Table 6 reports each item's discrimination (slopes on the general and specific factors) and the ordered intercepts $d_{jk}$ for the fitted bifactor model. By model definition, specific factor slopes are 0 for items assumed not measuring on that factor. Larger general slopes $a_{jg}$ indicate that the item is more sensitive to differences in the frailty score. In particular, item 3 (Toilet help) shows very large slopes, which likely reflects both the relatively low frequency of option 2 responses (Fig 1) and the fact that patients actually receiving help when using the toilet tend to either be high on the mobility dimension (reflecting low mobility), have high frailty, or both with very few exceptions. For interpretability along the general frailty dimension, we also report the general–axis thresholds $b_{jk}^{(g)} = -d_{jk}/a_{jg}$. Each $b_{jk}^{(g)}$ is the value of $\theta$ where the probability of responding at or above level $k$ equals 0.5 when the specific factors are

**Table 5. Model fit statistics for IRT models.**

| Model | RMSEA (90% CI) | SRMSR | TLI | CFI |
|---|---|---|---|---|
| Unidimensional | 0.1171 (0.1146, 0.1197) | 0.094 | 0.914 | 0.936 |
| Correlated-factor | 0.0588 (0.0562, 0.0614) | 0.054 | 0.978 | 0.984 |
| Bifactor | 0.0264 (0.0232, 0.0296) | 0.033 | 0.996 | 0.998 |

**Table 6. Bifactor GRM item parameters (intercept parameterization) and general–axis thresholds.** Values are estimates with SE in parentheses. $a_{jg}$ = general factor slope; $a_{js1}$ = Physical Functioning specific slope; $a_{js2}$ = Well-being/Mental Health specific slope. General–axis thresholds $b_{jk}^{(g)} = -d_{jk}/a_{jg}$ are computed with specific factors fixed at 0.

| Item | $a_{jg}$ | $a_{js1}$ | $a_{js2}$ | $d_{jk}$ | $b_{jk}^{(g)} = -d_{jk}/a_{jg}$ |
|---|---|---|---|---|---|
| 1 Return to life | 2.97 (0.08) | 0 | 1.14 (0.06) | {1.89, −2.23} (0.04, 0.05) | {−0.64, 0.75} |
| 2 Mobility | 5.14 (0.16) | 2.20 (0.15) | 0 | {−4.41, −7.72} (0.12, 0.18) | {0.86, 1.50} |
| 3 Toilet help | 11.51 (1.29) | 7.14 (0.91) | 0 | {−16.78} (1.87) | {1.46} |
| 4 Dressing help | 5.77 (0.24) | 3.31 (0.21) | 0 | {−7.12} (0.27) | {1.23} |
| 5 Dependent on support | 3.27 (0.07) | 1.21 (0.09) | 0 | {−0.28, −4.63} (0.04, 0.07) | {0.09, 1.42} |
| 6 Down/depressed/anx. | 1.40 (0.04) | 0 | 1.68 (0.06) | {−0.99} (0.03) | {0.71} |
| 7 General health | 2.43 (0.05) | 0 | 1.26 (0.06) | {2.99, −2.89, −6.11} (0.05, 0.05, 0.09) | {−1.23, 1.19, 2.51} |
| 8 Increased tiredness | 2.46 (0.09) | 0 | 2.60 (0.13) | {1.03} (0.05) | {−0.42} |
| 9 New pain | 1.08 (0.04) | 0 | 1.04 (0.05) | {−1.82} (0.03) | {1.69} |

Notes: For items with multiple ordered response levels, $d_{jk}$ and $b_{jk}^{(g)}$ are listed left-to-right in order of the thresholds. Items with $a_{jg} = 0$ would have no $b_{jk}^{(g)}$ (none here).

fixed at zero. Note that since $\theta$, $\theta_{s1}$ and $\theta_{s2}$ are assumed to be standard normal (mean 0, SD 1 in the reference groups), a patient at $\theta = 1$ is roughly one standard deviation frailer than the sample average. Items with $b^{(g)}$ close to a patient's $\theta$ are the most informative about that individual's position on the frailty continuum. The physical functioning items (2–5) are most informative from moderate to high frailty. The other items provide information throughout the entire score scale but mainly in the mid to upper score range.

## Score fairness

The expected item scores (ranging from 0 to the number of possible item responses) as functions of $\theta$ for each demographic group (age, education and sex) are shown in Fig 2. The items for which the curves differ between groups were identified as DIF items by the drop-sequential DIF analysis, while the others were used as anchor items for model identification. Visual inspection already suggests that most items behave similarly between groups. Using ESSD as an effect size metric (Table 7), items that exceed the 'small' DIF threshold of |ESSD| > 0.20 are found mainly for the age variable. Exceptions are item 8 for primary vs. university education and item 6 for sex with |ESSD| values barely exceeding 0.2. Item 6 and item 9 show medium DIF for age with |ESSD| > 0.50. Overall, the questionnaire demonstrates measurement invariance across sex and education, but small-to-moderate age-related bias on items 1, 6, 8 and 9.

Table 8 presents the subgroup-specific latent means and variances after anchoring on the items without DIF. Note that the score scale is assumed standard normal (in the reference groups) by assumptions of the model fitting algorithm. Thus, it theoretically ranges from $-\infty$ to $+\infty$, but our estimated values are all within the range (–2.6, 2.6). Respondents older than 76 years exhibited a mean $\hat{\theta}$ that was 0.45 standard deviations higher than those 76 or younger, and females had a mean $\hat{\theta}$ 0.28 standard deviations higher than males—both indicating worse recovery/well-being in these groups. In contrast, respondents with secondary or university education had mean $\hat{\theta}$ values 0.17 and 0.29 standard deviations lower than those with only primary education, respectively, suggesting comparatively better recovery, likely confounded by

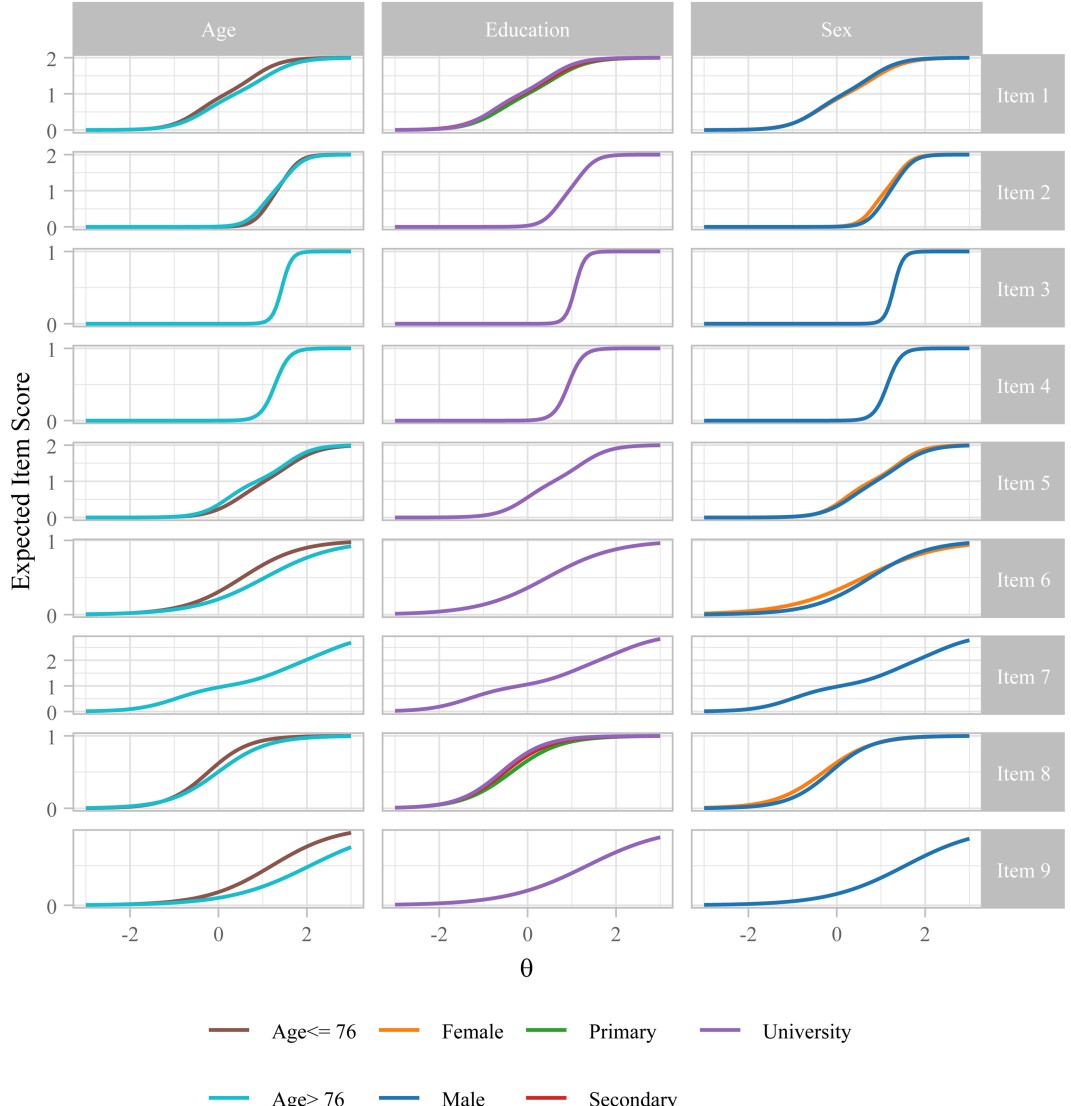

**Fig 2**. **Expected item scores vs. frailty score (θ) by age group, sex and education.** The expected item score reflects a weighted average of possible responses based on their estimated probabilities.

**Table 7**. ESSD for each item across grouping variables.

| Variable | Item 1 | Item 2 | Item 3 | Item 4 | Item 5 | Item 6 | Item 7 | Item 8 | Item 9 |
|---|---|---|---|---|---|---|---|---|---|
| Age | −0.24 | 0.07 | – | – | 0.20 | −0.53 | – | −0.28 | −0.75 |
| Education: Primary vs Secondary | 0.11 | - | - | - | - | - | - | 0.16 | - |
| Education: Primary vs University | 0.15 | – | – | – | – | – | – | 0.25 | – |
| Education: Secondary vs University | 0.04 | - | - | - | - | - | - | 0.09 | - |
| Sex | −0.08 | 0.11 | – | – | 0.10 | 0.26 | – | 0.13 | – |

Items with missing entries (-) correspond to anchor items for the given variable chosen by the employed iterative drop-sequential anchor-purification procedure. Thus, they are constrained to zero DIF by specification.

**Table 8**. Estimated latent means and variances (±95% CI) for all groups.

| Grouping variable | Group | Mean (95% CI) | Variance (95% CI) |
|---|---|---|---|
| Age | Age ≤ 76 | 0 (NA, NA) | 1 (NA, NA) |
| | Age > 76 | 0.45 (0.42, 0.49) | 0.90 (0.84, 0.96) |
| Sex | Male | 0 (NA, NA) | 1 (NA, NA) |
| | Female | 0.28 (0.25, 0.32) | 0.79 (0.74, 0.84) |
| Education | Primary | 0 (NA, NA) | 1 (NA, NA) |
| | Secondary | −0.17 (−0.21, −0.13) | 1.01 (0.94, 1.08) |
| | University | −0.29 (−0.34, −0.25) | 0.98 (0.90, 1.05) |

other variables. Estimated variances were similar within subgroups of education level, but slightly lower for females and people over 76 years of age compared to their counterparts.

### External validity

Fig 3 displays Kaplan–Meier survival curves stratified by quartile of the unidimensional and bifactor model $\hat{\theta}$ scores. There is a clear, monotonic decline in survival probability from Q1 (lowest $\hat{\theta}$) through Q4 (highest $\hat{\theta}$) over the 1,100-day follow-up after questionnaire completion. The survival probability is similar between the Q1 and Q2 groups, but decreases more rapidly for the subsequent quartiles, and the patterns were nearly identical for the unidimensional and bifactor-derived scores. The unidimensional $\hat{\theta}$ quartiles used for survival analysis corresponded to scores of: Q1 (<–0.44), Q2 (–0.44 to 0.23), Q3 ([0.23 to 0.85), and Q4 (>0.85). The bifactor $\hat{\theta}$ quartiles were: Q1 (<–0.71), Q2 (–0.71 to 0.04), Q3 ([0.04 to 0.57), and Q4 (>0.57). One should note that these scores are unitless and not directly comparable. A unidimensional

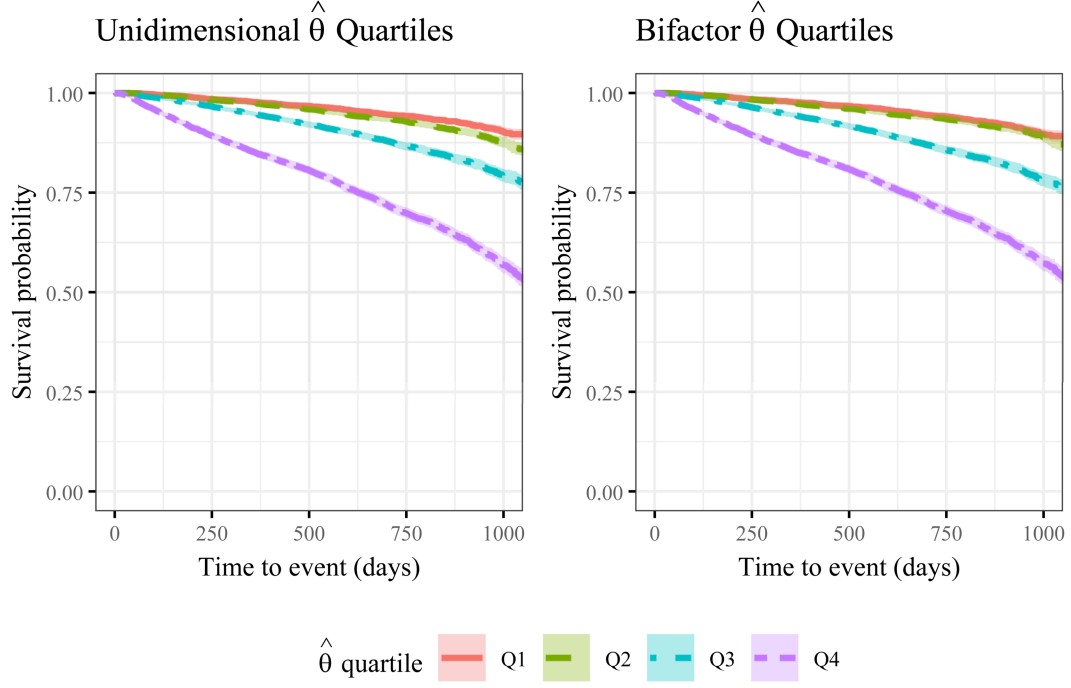

**Fig 3**. **Kaplan–Meier survival curves stratified by $\hat{\theta}$ quartile.** Quartiles Q1–Q4 correspond to increasing $\hat{\theta}$ (worse recovery); shaded bands are 95% CIs and vertical lines are events (death).

model score represents a composite of the general trait mixed with specific factors, whereas the bifactor score mathematically separates these components to measure a more "purified" general trait.

In Cox regression analyses, both scores were strongly associated with all-cause mortality. In models adjusted for age, sex, and education, each one-unit increase in $\hat{\theta}$ corresponded to an approximately 80% increase in the hazard of death for a 74.12 year old (average age in the dataset) male with primary school education (unidimensional: HR = 1.76, 95% CI: 1.62–1.90; bifactor: HR = 1.83, 95% CI: 1.68–1.99; Table 9). Results were consistent across the two scoring approaches, reflecting their high correlation ($r = 0.98$). The models indicated modest evidence of effect modification (Table 9). The association between frailty and mortality was somewhat stronger in females than in males (e.g., bifactor: HR for $\hat{\theta} \times$ female = 1.13, 95% CI: 1.04–1.24) and among respondents with university education compared to those with primary education (HR for $\hat{\theta} \times$ university = 1.18, 95% CI: 1.05–1.33). Interaction with age was negligible. These differences did not materially alter the overall conclusion that higher frailty scores were associated with increased mortality risk. Note that relatively small effects are also significant due to the large sample size. We evaluated the linearity assumption for the continuous predictors, age and $\theta$, by contrasting spline-based Cox models with models using linear terms. As no substantial deviations from linearity were observed, we retained the linear specification for ease of interpretation.

## Discussion

This study shows that a psychometrically robust, fair, and prognostically valid single-number frailty score can be derived from PROMs three months post stroke. Using MSA, EFA, and bifactor IRT modeling, we identified a dominant general frailty dimension alongside two clinically relevant subdomains, Physical Functioning and Well-being/Mental Health. DIF analyses demonstrated broad measurement invariance across sex and education, with modest age-related item bias. Importantly, the score exhibited strong external validity: Kaplan–Meier curves showed monotonic separation by frailty quartiles (Fig 3), and in Cox models adjusted for age, sex, and education, a higher frailty score was associated with substantially increased risk of death.

Our findings extend previous work on the measurement of frailty in stroke. Traditional approaches such as the Fried phenotype, Frailty Index, or administrative tools such as eFI and HFRS are largely rated by clinicians or based on coded

**Table 9. Cox regression modeling hazard of death utilizing frailty ($\theta$) scores from unidimensional and bifactor (general-factor $\theta$ scores) IRT models are shown in their respective columns.** Age and $\theta$ scores were centered before fitting the Cox models for easier main effect interpretation.

| Characteristic | Unidimensional model $\theta$ | | | Bifactor model $\theta$ | | |
|---|---|---|---|---|---|---|
| | HR | 95% CI | p-value | HR | 95% CI | p-value |
| $\theta$ | 1.76 | 1.62, 1.90 | <0.001 | 1.83 | 1.68, 1.99 | <0.001 |
| Age (years) | 1.07 | 1.07, 1.08 | <0.001 | 1.07 | 1.07, 1.08 | <0.001 |
| Sex | | | | | | |
| Male | — | — | | — | — | |
| Female | 0.67 | 0.62, 0.74 | <0.001 | 0.69 | 0.63, 0.75 | <0.001 |
| Education | | | | | | |
| Primary | — | — | | — | — | |
| Secondary | 0.83 | 0.75, 0.91 | <0.001 | 0.83 | 0.76, 0.92 | <0.001 |
| University | 0.70 | 0.63, 0.79 | <0.001 | 0.70 | 0.63, 0.79 | <0.001 |
| $\theta *$ Age (years) | 1.00 | 0.99, 1.00 | 0.3 | 1.0 | 0.99, 1.00 | 0.035 |
| $\theta *$ Sex | | | | | | |
| $\theta *$ Female | 1.15 | 1.06, 1.25 | 0.001 | 1.13 | 1.04, 1.24 | 0.005 |
| $\theta *$ Education | | | | | | |
| $\theta *$ Secondary | 1.08 | 0.98, 1.18 | 0.10 | 1.08 | 0.98, 1.19 | 0.10 |
| $\theta *$ University | 1.17 | 1.05, 1.31 | 0.006 | 1.18 | 1.05, 1.33 | 0.005 |

Abbreviations: CI = Confidence Interval, HR = Hazard Ratio.

data [14–16]. By contrast, the present score is derived directly from PROMs already collected at scale in Riksstroke, making measurement highly efficient in terms of cost and time. Previous studies of PROMs in stroke have applied IRT to new scale development [20], examined multidimensional structures [37], and shown that PROMs predict future use of healthcare care [21]. Our work adds to this literature by developing a frailty score, testing its fairness, and validating its prognostic value within a national registry.

Several limitations merit consideration. First, we did not compare our score with established frailty instruments, which limits direct comparability between measurement approaches and prevents the definition of specific clinical cut-offs (e.g., for mild, moderate, or severe frailty) without further calibration against these standards. Second, although DIF was generally small, some age-related differences were observed, particularly for mood and pain items; these may reflect true age-related vulnerability or attribution biases. Third, a notable exclusion from our score is a formal assessment of cognitive impairment. Although Riksstroke includes a screening item for memory and concentration, we omitted it to ensure a more homogeneous physical-physiological frailty construct. Cognition is undoubtedly a component of frailty in a broader sense, but including a single self-reported cognitive item could have compromised the unidimensionality of the scale. Future studies should evaluate how our physical frailty score interacts with objective cognitive screens. Lastly, "don't know" responses were treated as missing, which may not fully capture patient uncertainty.

The strengths of this study include the large and nationally representative sample and the rigorous psychometric evaluation. Because Riksstroke captures nearly all patients hospitalized with acute stroke in Sweden, the findings are likely to be generalizable to similar populations in other high-income settings with comprehensive registry infrastructures. Nevertheless, cultural and health system differences may influence item interpretation and score performance; replication in other contexts is needed.

To maximize the clinical utility of these findings, the strong association between frailty quartiles and survival (Fig 3) suggests this score could form the basis for future risk stratification tools. Validated risk tables estimating, for example, 3-year survival probabilities—could eventually be integrated into the registry feedback to aid clinicians and families in long-term care planning. Moreover, because this assessment captures patient-reported deficits in mood and fatigue that are often overlooked in brief clinical encounters, a high frailty score could serve as a 'red flag' to, for example, trigger a Comprehensive Geriatric Assessment or a review of home-care needs. Future studies should specifically link this score to healthcare utilization data, such as hospital readmissions and institutionalization, to further validate its value in guiding resource allocation.

In summary, we developed and validated a frailty score from PROMs in the Riksstroke three-month follow-up questionnaire. The score reflects a strong general frailty dimension, demonstrates fairness across demographic groups, and is associated with mortality. Beyond this specific setting, our findings illustrate how routinely collected PROM data can be leveraged to construct psychometrically sound and prognostically meaningful frailty measures. Such approaches may complement or substitute clinician-rated instruments, offering scalable ways to monitor recovery and guide quality improvement in stroke care. Future studies should test the generalizability of this strategy across different health systems, cultures, and measurement contexts, and benchmark PROM-derived frailty scores against established instruments.

## Acknowledgments

The authors are grateful to Riksstroke and all the participating hospitals.

## Author contributions

**Conceptualization:** Joakim Wallmark, Marie Eriksson.

**Data curation:** Joakim Wallmark, Marie Eriksson.

**Formal analysis:** Joakim Wallmark.

**Funding acquisition:** Marie Wiberg, Marie Eriksson.

**Methodology:** Joakim Wallmark, Marie Wiberg, Marie Eriksson.

**Project administration:** Marie Wiberg, Marie Eriksson.

**Software:** Joakim Wallmark.

**Supervision:** Marie Wiberg, Marie Eriksson.

**Validation:** Joakim Wallmark.

**Visualization:** Joakim Wallmark.

**Writing – original draft:** Joakim Wallmark.

**Writing – review & editing:** Joakim Wallmark, Marie Wiberg, Marie Eriksson.

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
