## [Decision Letter · Decision Letter 0]

21 Jan 2026

PONE-D-25-53227Developing and validating a frailty score based on patient-reported outcome 3 months after stroke: A Riksstroke based studyPLOS One

Dear Dr. Wallmark,

Thank you for submitting your manuscript to PLOS ONE. After careful consideration, we feel that it has merit but does not fully meet PLOS ONE’s publication criteria as it currently stands. Therefore, we invite you to submit a revised version of the manuscript that addresses the points raised during the review process.

**ACADEMIC EDITOR:**

This manuscript reports on the development and validation of a frailty score based on patient-reported outcomes at three months after stroke and has been reviewed by two expert reviewers. Both the methodology and the discussion are considered appropriate, and the study is regarded as meaningful and of clinical significance. Before acceptance, the authors are required to address the questions raised by one of the reviewers.

We look forward to receiving your revised manuscript.

Kind regards,

Yoshitaka Ishibashi

Academic Editor

PLOS One

Journal Requirements:

2. For studies involving third-party data, we encourage authors to share any data specific to their analyses that they can legally distribute. PLOS recognizes, however, that authors may be using third-party data they do not have the rights to share. When third-party data cannot be publicly shared, authors must provide all information necessary for interested researchers to apply to gain access to the data. (https://journals.plos.org/plosone/s/data-availability#loc-acceptable-data-access-restrictions)

Reviewers' comments:

Reviewer's Responses to Questions

**Comments to the Author**

1. Is the manuscript technically sound, and do the data support the conclusions?

Reviewer #1: Yes

Reviewer #2: Yes

2. Has the statistical analysis been performed appropriately and rigorously?

Reviewer #1: I Don't Know

Reviewer #2: I Don't Know

3. Have the authors made all data underlying the findings in their manuscript fully available?

Reviewer #1: No

Reviewer #2: Yes

4. Is the manuscript presented in an intelligible fashion and written in standard English?

Reviewer #1: Yes

Reviewer #2: Yes

5. Review Comments to the Author

Reviewer #1: This is an interesting study on a novel concept of using patient reported status to develop a frailty measure.

Minor comments

page 2/15 Introduction

Lines 32 to 42 - This paragraph repeats a significant amount of information in the paragraph immediately above. It appears to be an oversight on the authors part and needs to be consolidated.

Line 44 - Please list the original reference where the Riksstroke questionnnaire was derived/validated and an internet link to the questionnaire. Why did the authors choose only 9 out of 12 questions from the original patient Rikkstroke questionnaire? why were the questions on a) transportation, b) thoughts, concentration and memory and their effects on ability to carry out daily activities omitted? Impaired cognition is one of several major geriatric syndromes that contribute to frailty. (Edmonton Frail Scale and Frailty Index includes cognition). The authors should justify the omissions especially on the part on cognition.

page 6/15

Can the authors describe the basic or simple aspects of the results further. For example , what is the range of possible scores. (minimum to maximum) of this PROM scale? What is the score range of the first and second quartile (which seem to have similar mortality rates), and score range of the third and fourth quartile? Can any of these quartiles be equated to no frailty, mild, moderate and severe frailty? If this cannot be done for the time being and needs further validation studies against standardised tools, then the authors can explain as such.

In the discussion section, can the authors add in discussion about the potential future clinical utility of this frailty measurement tool? Eg a risk stratification table to estimate 3 year survival rate perhaps? And future studies to correlate with care needs and health care resources utility. Many readers will be clinicians who have some basic statistical knowledge but the main points of interest would be the daily use of this frailty assessment tool in the care plan of the person living with stroke.

page 8/15

Table 5

Are the terms two-factor and bifactor standardised statistical terms? Although they are explained further on in the manuscript, it can be confusing to the reader as they are close in meaning in English. If possible the authors can consider changing one of the terms eg two-factor

page 11/15

Line 286 - .....1 100-day follow up....., do the authors mean the first 100 days follow up?

Reviewer #2: I think the authors have thoroughly assessed and validated a frailty score based on patient reporte data. Based on my limited experience in this field of prediction score creation, this study sounds safe to be accepted.

6. PLOS authors have the option to publish the peer review history of their article (what does this mean?). If published, this will include your full peer review and any attached files.

Reviewer #1: No

Reviewer #2: No

---

## [Author Response · Author response to Decision Letter 1]

29 Jan 2026

See the uploaded Response to Reviewers pdf file.

---

## [Editor Report · Decision Letter 1]

4 Feb 2026

Developing and validating a frailty score based on patient-reported outcome 3 months after stroke: A Riksstroke based study

PONE-D-25-53227R1

Dear Dr. Wallmark,

We’re pleased to inform you that your manuscript has been judged scientifically suitable for publication and will be formally accepted for publication once it meets all outstanding technical requirements.

Kind regards,

Yoshitaka Ishibashi

Academic Editor

PLOS One
---

## [Editor Report · Acceptance letter]

PONE-D-25-53227R1

PLOS One

Dear Dr. Wallmark,

I'm pleased to inform you that your manuscript has been deemed suitable for publication in PLOS One. Congratulations! Your manuscript is now being handed over to our production team.

Kind regards,

on behalf of

Dr. Yoshitaka Ishibashi

Academic Editor

PLOS One